# FSL-QuickBoost: Minimal-Cost Ensemble for Few-Shot Learning

Yunwei Bai
baiyunwei@u.nus.edu
National University of Singapore

Bill Yang Cai
billcai@alum.mit.edu
Amazon Web Services

Ying Kiat Tan
yingkiat@u.nus.edu
National University of Singapore

Zangwei Zheng
zangwei@u.nus.edu
National University of Singapore

Shiming Chen
gchenshiming@gmail.com
Mohamed bin Zayed University of
Artificial Intelligence

Tsuhan Chen
tsuhan@nus.edu.sg
National University of Singapore

## ABSTRACT

Few-shot learning (FSL) usually trains models on data from one set of classes, but tests them on data from a different set of classes, providing a few labeled support samples of the unseen classes as a reference for the trained model. Due to the lack of training data relevant to the target, there is usually high generalization error with respect to the test classes. Some existing methods attempt to address this generalization issue through ensemble. However, current ensemble-based FSL methods can be computationally expensive. In this work, we conduct empirical explorations and propose an ensemble method (namely QuickBoost), which is efficient and effective for improving the generalization of FSL. Specifically, QuickBoost includes an alternative-architecture pretrained encoder with a one-vs-all binary classifier (namely FSL-Forest) based on random forest algorithm, and is ensembled with the off-the-shelf FSL models via logit-level averaging. Extensive experiments on three benchmarks demonstrate that our method achieves state-of-the-art performance with good efficiency.

## CCS CONCEPTS

• **Computing methodologies → Ensemble methods**.

## KEYWORDS

Few-Shot Classification, Ensemble, Deep Learning, Machine Learning, Computation Efficiency

**ACM Reference Format:**
Yunwei Bai, Bill Yang Cai, Ying Kiat Tan, Zangwei Zheng, Shiming Chen, and Tsuhan Chen. 2024. FSL-QuickBoost: Minimal-Cost Ensemble for Few-Shot Learning. In *Proceedings of the 32nd ACM International Conference on Multimedia (MM '24), October 28-November 1, 2024, Melbourne, VIC, AustraliaProceedings of the 32nd ACM International Conference on Multimedia (MM'24), October 28-November 1, 2024, Melbourne, Australia.* ACM, New York, NY, USA, 10 pages. https://doi.org/10.1145/3664647.3681446

## 1 INTRODUCTION

Few-shot learning (FSL) in the context of classification, is defined as a classification problem under a lack of supervision information

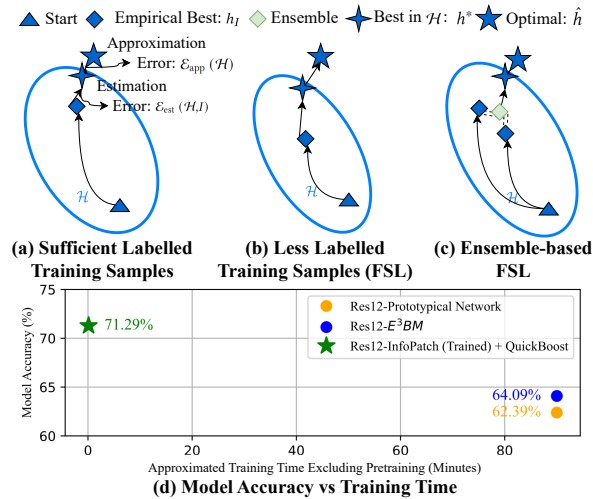

**(a) Sufficient Labelled Training Samples**  **(b) Less Labelled Training Samples (FSL)**  **(c) Ensemble-based FSL**

**(d) Model Accuracy vs Training Time**

Figure 1: Motivation illustration. (a) The estimation and approximation errors (Formula 9) are small under sufficient training with supervision. (b) The errors becomes larger with less relevant supervision in FSL. (c) Ensemble has lower errors compared to (b), despite less supervision under the FSL setting. (d) Compared to other ensemble-based FSL methods (i.e., $E^3BM$ [35]), our QuickBoost achieves state-of-the-art performance at minimal cost.

on the target task [48, 55]. In many real-world scenarios, sufficient labeled datasets are usually not readily available [19, 55, 61], especially for rare object types. Therefore, the few-shot learning classification algorithms serve to overcome the problem of generalization in the face of such data constraints. It is commonly assumed that FSL models undergo training based on data from a different set of classes compared to those of the test data set [3, 50, 55, 59]. With few-shot training, models can predict classes of data they have not been trained on.

However, the data constraint also limits the FSL model performance. In deep neural networks (DNN), lower generalization error is generally achieved by having a large number of labelled training examples coupled with models with sufficient parameters [29, 39, 55]. With less supervision information, local search (e.g., gradient descent) tends to get stuck at local minima [62]. Meanwhile, due to the lack of relevant data, the best hypothesis in the

hypothesis space* may not represent the target well [62]. Furthermore, on the one hand, a single model with less parameters can be limited in its expressiveness to represent the target accurately; on the other hand, with limited data, models with more parameters can overfit [39], composing a dilemma for FSL.

To tackle the high generalization error, or the core challenge, in FSL classification, different categories of methods are proposed. Notably, in the *data aspect*, data augmentation techniques are used to augment the training dataset for FSL models, providing more supervision based on datasets related to the target tasks [21, 26, 37, 46, 52, 56]. In the *model aspect*, representation learning techniques project training data to lower dimension, setting the similar samples closer together while the dissimilar further apart [12, 34, 48, 50, 53, 59]. Both the data augmentation and the embedding learning techniques tackle the FSL problems mainly via integrating more prior information to FSL models [55]. In the *algorithm aspect*, meta-learning algorithms [4, 32, 45, 49] can learn good model parameter initialization which enables quick adaptation to the new unseen task. However, the augmentation can introduce bias as argued in works like Xu and Le [57]. Meanwhile, model-based embedding-learning approaches are limited in performance, while algorithm-based meta-learning algorithms can be unstable to train with nested training loops [2].

To analyse the problem further, we define a hypothesis space $\mathcal{H}$. We have $\hat{h}$ as the most ideal optimal classifier in theory, $h^*$ as the best classifier within the hypothesis space considered, and $h_I$ as the classifier we obtain via thorough empirical risk minimizing over available training data. The expected difference between the risk of $h_I$ and $\hat{h}$ in FSL (i.e., the best room of improvement that can be achieved in theory) can be decomposed into approximation error and estimation error, as illustrated in Formula 9 and Figure 1. When there is sufficient and relevant supervision data, both estimation error (distance between $h^*$ and $h_I$) and approximation error (distance between $\hat{h}$ and $h^*$) are relatively small, as illustrated in Figure 1(a). When the relevant labelled data reduce, both of the errors increase. In particular, the estimation error increases due to a higher chance of the estimator $h_I$ getting stuck at local minima during the optimum search. The approximation error increases due to a decreased amount of labelled data learnt relevant to the target; without enough relevant data, the best target of search $h^*$ becomes relatively inaccurate, as illustrated in Figure 1(b). Ensemble mitigates these two errors. Specifically, despite accessing less labelled training data, the estimation error can be reduced by considering alternative local search results which average out bias of local minima. Meanwhile, the approximation error can be reduced through greater model expressiveness; as ensemble involves more model parameters, it can consider a larger hypothesis space and thus more accurate targets of search [39, 55, 62], as illustrated in Figure 1(c).

According to existing literature, plenty of empirical and theoretical results have demonstrated the effectiveness of ensemble [62], in tackling the approximation and estimation errors, which are the precise pain points of FSL. Meanwhile, "how to ensemble effectively" is a question equally worthy of consideration. In fact, ensemble has

certain preconditions to fulfill in order to unleash and maximize its potential benefits. It also has disadvantages. Specifically, to achieve good ensemble performance, the predictors have to possess a fair accuracy on their own and enough difference against other predictors to achieve complementary difference in their predictions. The predictions can then complement one another well to achieve improved overall accuracy [16, 62]. Besides, as multiple models need to be trained, ensemble can be disadvantageous in terms of computation efficiency and implementation convenience, which are relevant considerations for real-world applications [15, 28, 62].

Therefore, our goal is to reduce generalization error in trained FSL models, boosting their performance via resource-efficient ensemble. In this paper, we design an ensemble method for FSL, called *FSL-Quickboost* (Quickboost for short). Essentially, QuickBoost involves pretrained features from an alternative encoder and a one-vs-all [39] binary classifier to be ensembled with trained FSL models. The binary classifier, called *FSL-Forest*, is random-forest-based [5]. We choose the random-forest algorithm due to its widely demonstrated decency in performance. Importantly, it is ready for few-shot classification tasks at minimal cost. To implement the FSL-Forest classifier, we directly use the pretrained encoder to produce image features, whose pairwise feature difference are fed as the input features to FSL-Forest. The extensive experiments on standard benchmarks demonstrate the superior performance.

Our main contributions can be summarized as follows:

- We analyse why ensemble can be a simple-yet-effective solution to the challenging set-ups in FSL.
- We propose an ensemble scheme for trained FSL models, which involves a pretrained encoder paired with a random-forest-based FSL classifier. The classifier predicts based on the average among a set of decision tree stumps, each of which makes predictions based on pairwise differences among pretrained data features. The decision trees consider the elementwise value differences of each feature pair, strengthening initial predictions of the original FSL model.
- We provide extensive experiments to show the advantages and potential of FSL-QuickBoost. For example, our FSL-QuickBoost obtains approximately 6%, 6% and 7% performance improvement in 5-way-5-shot tasks on *tiered*Imagenet [44], *mini*Imagenet [44] and Cifar-FS [30], respectively.

## 2 RELATED WORKS

**Few Shot Learning.** Few-shot learning classification algorithms can be broadly categorized into data-based approach, algorithm-based approach and model-based approach [56]. Among *data-based algorithms*, Gao et al. [21], Hariharan and Girshick [26], Mishra et al. [37], Schwartz et al. [46], Verma et al. [52], Wang et al. [56] expands the training dataset by, for example, combining object characteristics across base classes, but the generation may involve bias since certain combinations can be invalid [57]. Among *algorithm-based* approaches, meta-learning [4, 18, 32, 45, 49] learns a good initialization of model parameters, and *model-based* approaches mainly involve metric-based learning, where models learn an embedding space that contrasts input samples at a lower dimension [10, 11, 34, 48, 50, 53, 59]. Most of the methods are purely deep-learning-based, which are data-hungry and time-consuming to train [54]. In particular, meta-learning algorithms like MAML [18] can

---

*Hypothesis space size is defined as the number of hypotheses. A hypothesis is a way of classification that best classifies the training data seen so far, given the representational ability of the function. In general, hypothesis space grows when there are more training data or model parameters [39].

be unstable to train [2]. Moreover, due to the challenging nature of FSL, most methods have limited performance. Therefore, we are motivated to improve the performance of existing FSL algorithms while keeping the ensemble procedure simplistic.

In this paper, we examine a few typical FSL models to be ensembled with our methods, or namely the Prototypical Network (ProtoNet), Matching Network (MatchNet), Relation Network (RelationNet), FEAT, DeppSet and Infopatch. The FSL model encoders include the Conv4, Res12 or Res18 structures. The Conv4 encoder [50, 59] is a relatively simple 4-layer convolutional neural network (CNN) [31]. Res12 and Re18 [59] are of a more complex Resnet [27] structure. The Prototypical Network [48] is similar to the Matching Network [53]. Both of these models produce embeddings for pairs of support set sample and query, before measuring their similarity scores using either the euclidean distance or cosine similarity. When there is just one shot for the support set, the Matching Network and Prototypical Network are equivalent; when there are multiple shots, the Prototypical Network considers the average embedding of all support set samples within one class, while the Matching Network considers the support samples individually. The Relation Network [50] is similar, except that the algorithm uses a neural network to calculate the degree of similarity between two image embeddings. FEAT [59], DeepSet [59] and InfoPatch [34] have a similar set-up, but FEAT and DeepSet add in transformative layers (e.g., self-attention-based mechanism [51]) to produce more flexible embedding. InfoPatch tries to produce more accurate embeddings by blocking part of the query images, making the task harder [34].

**Ensemble.** Ensemble can be achieved using different approaches, including independently constructed ensembles and coordinated constructed ensembles [15, 28]. For an *independently constructed ensemble*, boosting ensemble method combines weak learners to produce a strong learner; random forest is also a type of independently constructed ensemble that combines the predictions of decision trees through majority voting [5]. The collection of tree structured classifiers can be decision trees [6, 7]. For *coordinated constructed ensemble*, Adaboost [15, 20] algorithm constructs new hypotheses incrementally. In the intersection field of FSL and model ensemble, Dvornik et al. [16] proposes to train multiple few-shot models with a loss function that encourages diversity via KL-divergence and cooperation via cosine similarity. Another work, Liu et al. [35] $E^3BM$ learns and combines an ensemble of epoch-wise Bayes models. There are also other works involving ensemble of features or classifiers for few-shot learning [1, 3, 9, 25, 33, 58]. The major disadvantage of these ensemble schemes lies in the extra computation overhead and complication in implementation introduced. Importantly, most of these methods (e.g., Dvornik et al. [16], Liu et al. [35]) involve training of ensemble itself. Orthogonal to these methods, our design focuses on efficiency, enabling easy and quick performance boost via test-time-only ensemble, without involving any extra training of the original model.

## 3 METHOD

### 3.1 Problem Formulation

FSL classification tasks commonly follow an $N$-way-$K$-shot set-up; $N$ represents the number of classes being classified and $K$ denotes the number of labeled samples per class. These $N \times K$ samples,

denoted as $S = \{x_1^{(1)}, x_2^{(1)}, \ldots, x_{K-1}^{(N)}, x_K^{(N)}\}$, are referred to as the *support set*. It is commonly assumed that $h$ is trained on a set of base classes $C^b$ with complete supervision information, and then tested on a disjoint set of novel classes $C^b$ (i.e., $C^b \cap C^n = \emptyset$). $\forall c \in C^n$, $c$ contains limited supervision information, or just a few labeled samples. These few labeled samples serve as the support set $S$ during testing [14, 48, 50, 55, 59].

During *training*, an FSL classifier $h$ is presented with a labelled support set comprising samples from $N$ classes, and a query $q$ sub-sampled from the $N$ classes, and $h$ has to learn or predict which category $q$ belongs to by referring to $S$. The $N$ classes are a subset of $C^b$. During *testing*, $h$ has to predict classes of $q$ based on labels of $S$, where both $q$ and $S$ are sampled from $C^n$. Note that according to inductive FSL set-up [3], the support set sample labels from $C^n$ are not used during training.

### 3.2 FSL-QuickBoost

**Overview.** There are two important characteristics of ensemble. *Firstly*, ensemble consumes extra computation resources, which needs to be considered in practical applications. *Secondly*, in general, to maximize the effectiveness of ensemble, ensembled models need to be as accurate as possible individually, while possessing differences in logit output against one another's, so as to achieve a complementary effect. As such, it is desirable if we can have ensemble which achieves both *computation efficiency* and the *complementary difference*. Therefore, guided by these principles, we aim to develop an effective ensemble scheme for FSL, which boosts single classifiers' performance at minimal cost, thus minimizing the pain points of both FSL and ensemble.

We introduce the FSL-QuickBoost ensemble scheme, where we restrain our scope to just one FSL-Forest model $h_2$ for ensemble with a trained FSL model $h_1$. $h_1$ comprises an encoder $\Phi_1$ and a classifier $t_1$, and $h_2$ also has an encoder $\Phi_2$ and a classifier $t_2$. Essentially, we try to achieve the aforementioned complementary difference at both feature level and classifier level. At the *feature level*, we adopt different encoders for $h_1$ and $h_2$ respectively. At the *classifier level*, we apply logit weighted-averaging of a random-forest-based [5] one-vs-all [39] binary classifier ($t_2$) output and the original FSL classifier ($t_1$) output. Meanwhile, for $t_2$, the training data required is few, and the training gets completed fast. The overall architecture of QuickBoost is presented in Figure 2, which includes four stages: stage (a) Pretraining of Encoder, stage (b) Preparation of Dataset, stage (c) FSL-Forest and stage (d) Ensemble. In stage (a), we conduct pretraining of encoder $\Phi_2$ on base classes $C^b$. For stage (b), we conduct preparation of dataset $D$ based on the $\Phi_2$ encoder. For stage (c), we conduct training of FSL-Forest classifier $t_2$, based on the dataset $D$. For stage (d), we conduct ensemble of $h_2 = t_2 \circ \Phi_2$ and a trained FSL model $h_1$ via weighted logit averaging.

**(a) Pretraining of Encoder.** Some works propose that the mainstream meta-learning algorithms in FSL are sub-optimal [14, 23, 40, 41], and they demonstrate that a fine-tuned pretrained encoder can achieve equivalent or even more superior performance as complex meta-learning models [13, 14, 23, 41]. Essentially, while simple, pretrained features can be effective for FSL classification. Following the practise, as the first step, we pretrain an encoder $\Phi_2$ on base classes $C^b$, via standard cross entropy loss minimization [14].

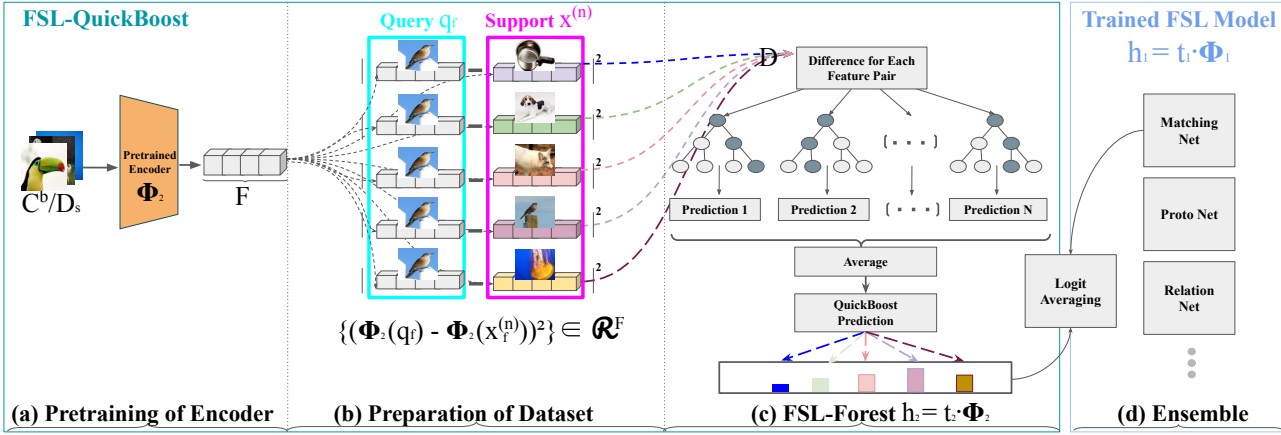

**Figure 2: The overall architecture of our FSL-QuickBoost ensemble. In stage (a), an encoder is pretrained on base classes. In stage (b), the squared of pairwise feature differences of sub-sampled images, along with labels indicating whether the image pairs are of the same class, are stored as a training dataset file. In stage (c), a one-vs-all binary classifier based on random forest is trained on the stage (b) dataset. In stage (d), weighted-averaging is performed among the random forest classifier and a model trained via standard FSL algorithms like the Relation Net.**

Specifically, given a dataset $D_s$ consisting of data from base classes $C^b$, we obtain the model parameters $\theta^*$ for an encoder $\Phi_2^{\theta^*}$:

$$\theta^*(D_s) = \arg\min_\theta \frac{1}{|D_s|} \sum_{(x,y)\in D_s} -\log p_\theta(y \mid x). \quad (1)$$

We use $\Phi_2^{\theta^*}$ ($\Phi_2$ in short) for the subsequent preparation of dataset.
**(b) Preparation of Dataset.** After obtaining $\Phi_2$, we sub-sample positive and negative image pairs $\{x_1, x_2\}$ from the base classes $C^b$, and produce pairwise supervision labels as:

$$\boldsymbol{y} = \mathbb{1}(y_1 = y_2). \quad (2)$$

For each pair $\{x_1, x_2\}$, where $x_i$ is $\in \mathbb{R}^F$, we calculate the squared elementwise feature difference, denoted as $\text{diff}(x_1, x_2)$. In the context of $N$-way-$K$-shot tasks, it is:

$$\{(\Phi_2(q_f) - \Phi_2(x_f^{(n)}))^2\} \in \mathbb{R}^F, \quad (3)$$

for $n \in [1, N]$ and $f \in [1, F]$. Note that $F$ is the total number of channels in $\Phi_2(x)$, and $q$ is the query.

We save tuples $D := \{(\text{diff}(x_i, x_j), \boldsymbol{y})\}$ for $i, j \in [1, M]$, where $M$ denotes the total number of samples collected from base classes $C^b$. These tuples are stored as data files for subsequent training of the random forest classifier $t_2$.
**(c) FSL-Forest.** We perform training on a random forest classifier $t_2$ with the prepared input $D$. The random forest classifier is named "FSL-Forest". The classifier constitutes a set of decision trees, each taking in the squared feature difference and returning a similarity relation score for the input image pair. Each decision tree partitions the input features recursively, based on some threshold, such that samples with the same labels or target values are grouped together. This procedure is also known as "impurity minimization".

Let data at node $m$ of the decision tree be denoted as $Q_m$ with $N_m$ samples. Here, each element of $Q_m$, $d$, is each element-wise squared difference of two input vectors, originated from the pair

of images for comparison. Let $\bar{d}_m = \frac{1}{N_m}\sum_{d\in Q_m} d$. The "impurity" $H(\cdot)$ is defined as the following mean squared error function [6–8]:

$$H(Q_m) = \frac{1}{N_m}\sum_{d\in Q_m}(d - \bar{d}_m)^2. \quad (4)$$

For each decision tree in the random forest classifier $t_2$, let $\omega$ denote the model parameters. Let $e_m$ be each candidate's split threshold value. Let threshold $e_m$ partition a node into two subsets, which are respectively denoted as $Q_m^{\text{left}}(\omega) = \{d|d < e_m, d \in Q_m\}$ and $Q_m^{\text{right}}(\omega) = Q_m \backslash Q_m^{\text{left}}(\omega)$. The training procedure of each decision tree is tantamount to the following equation [6–8]:

$$G(Q_m, \omega) = \frac{N_m^{\text{left}}}{N_m}H(Q_m^{\text{left}}(\omega)) + \frac{N_m^{\text{right}}}{N_m}H(Q_m^{\text{right}}(\omega)), \quad (5)$$

$$\omega^* = \arg\min_\omega G(Q_m, \omega). \quad (6)$$

Output scores $t_2(\cdot)$ from the random forest are based on the average of individual output of each decision tree parameterized by $\omega^*$, in a set of decision trees [6–8].

The rationale of the design is that each value of the squared feature difference represents the visual strength of a particular feature difference in the original data pair. If certain discrepancies corresponding to important features are strong enough, the data pair tends to be of different classes.
**(d) Ensemble.** Given two pretrained encoders $\Phi_1$ and $\Phi_2$, and two classifiers $t_1$ and $t_2$ for $h_1$ and $h_2$ respectively, assuming that the output from $h$ are normalized, ensemble returns $\alpha_1 \cdot h_1(q, S) + \alpha_2 \cdot h_2(q, S) = \alpha_1 \cdot t_1 \circ \Phi_1(q, S) + \alpha_2 \cdot t_2 \circ \Phi_2(q, S)$ at the logit level. Note that $q$ is the query, and $S$ is the average embedding of each support set class. Besides, $\alpha_1$ is the test-accuracy of $h_1$, and $\alpha_2$ is the test-accuracy of $h_2$ on few labelled support samples. Essentially, for ensemble with a high-accuracy FSL model, QuickBoost gives more priority to the original predictions. For an $N$-way-$K$-shot support set $S$ and a query $q$, $t_2(\cdot)$ input

is $\{\texttt{diff}(q, x_1^{(1)}), \texttt{diff}(q, x_2^{(1)}), \ldots, \texttt{diff}(q, x_{K-1}^{(N)}), \texttt{diff}(q, x_K^{(N)})\}$ for $x$ as a sample from one of the novel classes $C^n$. Here, $\texttt{diff}(\cdot, \cdot)$ denotes the set of element-wise squared difference between two features. The final ensemble prediction for each query corresponding to one support set is rendered as:

$$\arg\max_n \alpha_1 \cdot \text{norm}(h_1^n(\cdot)) + \alpha_2 \cdot \text{norm}(h_2^n(\cdot)) \quad \text{for } 1 \leq n \leq N, \ (7)$$

where the normalization function $\text{norm}(u)$ for a real value $u \in U$ and $U \in \mathbb{R}^N$ is defined as:

$$\text{norm}(u_i) = \frac{u_i - \min_j u_j}{\max_j u_j - \min_j u_j} \quad \text{for } 1 \leq i, j \leq N. \quad (8)$$

Note that the ensemble scheme is plug-and-play, happening only during inference stage, where the trained FSL model does not need to be trained further.

## 4 THEORETICAL INSIGHTS

In essence, FSL suffers from a high degree of approximation and estimation error, and ensemble can tackle such a pain point precisely. **Core Challenge(s) in FSL?** The core challenge is high difference between the best hypothesis obtainable via existing training data, and the best hypothesis in theory. Such a difference is mainly caused by the lack of supervision for the target tasks.

We define a hypothesis space $\mathcal{H}$, the expected risk $R(h)$ as $\int \ell(h(x), y) dp(x, y) = \mathbb{E}[\ell(h(x), y)]$. Empirical risk as $R_I(h)$ as $\frac{1}{I} \sum_{i=1}^{I} \ell(h(x_i), y_i)$, where $I$ is the total number of available training samples. Furthermore, we have $\hat{h} = \arg\min_h R(h)$ as the function that minimizes $R(h)$; $h^* = \arg\min_{h \in \mathcal{H}} R(h)$ as the function in $\mathcal{H}$ that minimizes $R(h)$; $h_I = \arg\min_{h \in \mathcal{H}} R_I(h)$ as the function in $\mathcal{H}$ that minimizes $R_I(h)$. Essentially, $\hat{h}$ is the most ideal optimal classifier, $h^*$ is the best classifier within the hypothesis space considered, and $h_I$ is the classifier we can obtain via thorough empirical risk minimizing training over available training data. The expected risk difference between $h_I$ and $\hat{h}$ can be written as:

$$\mathbb{E}\left[R(h_I) - R(\hat{h})\right] = \underbrace{\mathbb{E}\left[R(h^*) - R(\hat{h})\right]}_{\mathcal{E}_{\text{app}}(\mathcal{H})} + \underbrace{\mathbb{E}\left[R(h_I) - R(h^*)\right]}_{\mathcal{E}_{\text{est}}(\mathcal{H}, I)}. \quad (9)$$

Here, $\mathcal{E}_{\text{app}}(\mathcal{H}, I)$ is the approximation error, and $\mathcal{E}_{\text{est}}(\mathcal{H})$ is the estimation error [55].

The *approximation error*, representing the expected risk difference between $h^*$ and $\hat{h}$, arises when the optimal classifier does not exist within the considered hypothesis space. In deep learning, this error can be caused by both the lack of relevant supervision data and limited representational ability of deep neural networks. Generally, in FSL, as the target-task classes are novel, the chance of the optimal classifier existing further outside of the hypothesis space is high compared to standard classification problems [55]. This error is also known as the representational issue in Zhou [62].

The *estimation error* arises when there is a difference between the obtainable trained classifier and the best classifier within the hypothesis space considered [55]. This error can be caused by getting stuck at local minima during training, which relates to computational and statistical issues in Zhou [62]. Generally, the issues

can be mitigated with training on a large amount of labelled data [29, 39, 55]. FSL remains challenging given the lack of such access. **Why Ensemble?** In short, ensemble is a simple-yet-effective solution which tackles the aforementioned issues (i.e., representational, computational and statistical) precisely [62].

Given FSL models $\{h_I^1 \ldots h_I^T\}$, where $h_I(x)$ returns the logits of $x$, simple averaging gives the combined output $P(x)$ as:

$$P(x) = \frac{1}{T} \sum_{i=1}^{T} h_I^i(x). \quad (10)$$

Suppose the underlying true function to learn is $\hat{h}(x)$, and $x$ is sampled according to a distribution $p(x)$. The output of each learner can be written as the true value plus an error item $\epsilon$, which is a result of both $\mathcal{E}_{\text{app}}(\mathcal{H}, I)$ and $\mathcal{E}_{\text{est}}(\mathcal{H})$ i.e.,

$$h_I^i(x) = \hat{h}(x) + \epsilon_i(x), \quad i = 1, \ldots, T. \quad (11)$$

The mean squared error of $h_I^i$ can be written as:

$$\int \left(h_I^i(x) - \hat{h}(x)\right)^2 p(x) dx = \int \epsilon_i(x)^2 p(x) dx. \quad (12)$$

The averaged error made by the individual learners is:

$$\overline{\text{err}}(h_I) = \frac{1}{T} \sum_{i=1}^{T} \int \epsilon_i(x)^2 p(x) dx. \quad (13)$$

The expected error of the combined learner (i.e., ensemble) is:

$$\begin{aligned}
\text{err}(P) &= \int \left(\frac{1}{T} \sum_{i=1}^{T} h_I^i(x) - \hat{h}(x)\right)^2 p(x) dx \\
&= \int \left(\frac{1}{T} \sum_{i=1}^{T} \epsilon_i(x)\right)^2 p(x) dx.
\end{aligned} \quad (14)$$

If we assume that the errors $\epsilon_i$'s have zero mean and are uncorrelated, i.e.,

$$\int \epsilon_i(x) p(x) dx = 0 \text{ and } \int \epsilon_i(x) \epsilon_j(x) p(x) dx = 0, \quad (15)$$

for $i \neq j, j \in [1, T]$, we can have:

$$\text{err}(P) = \frac{1}{T} \overline{err}(h_I). \quad (16)$$

This suggests that ensemble error is smaller by a factor of $T$ than the averaged error of the individual learners [62]. When $\epsilon$ increases in magnitude, which is usually the case in FSL, $\overline{\text{err}}(h)$ increases, leading to a larger improvement brought by ensemble. From the perspective of approximation and estimation error, on the one hand, approximation error can be reduced via better representational ability of ensemble, which can cover a larger hypothesis space; on the other hand, estimation error characterized by local minima can be reduced through combining multiple local search results. The errors can be reduced best when ensemble involves accurate classifiers sharing little correlation (See assumption of Formula 15), which is also the complementary difference we refer to earlier.

# 5 EXPERIMENTS

## 5.1 Experimental Configurations

**Benchmark Datasets.** We use three popular FSL datasets, *mini*Imagenet, *tiered*Imagenet and Cifar-FS [49]. Both the *mini*Imagenet dataset and the *tiered*Imagenet are sampled from the imagenet dataset [44], and the Cifar-FS dataset are sampled from the Cifar-100 dataset [30]. In both the *mini*Imagenet dataset and the Cifar-FS dataset, there are 100 classes (train/validation/test classes = 64/16/20) [32]. Each of the class contains 600 images. In the *tiered*Imagenet dataset, there are more categories (train/validation/test classes = 351/97/160) [59].

**Implementation Details.** For pretraining of the feature encoders, we use a randomly initialized Res18 [27] model, whose last fully-connected layer. For the input data, we only apply normalization without any augmentation. We use a stochastic gradient descent (SGD) optimizer. The batch size is 128. The momentum is 0.9. The weight decay rate is 0.0001. The initial learning rate is 0.1. We train the model for 100 epochs, and the learning rate is reduced by a factor of 0.1 during the 30th, 60th and 90th epoch. The pretraining only involves the respective train split of the FSL datasets to prevent data leakage. After obtaining the image feature encoder for each of the dataset, we combine pairs of image feature vectors obtained via the pretrained encoder, and merge two 512-vectors into one 512-vector by computing their squared vector difference. Each merged 512-vector corresponds to either a positive label representing same-class pairs, or a negative label representing different-class pairs. We sample 12,800 pairs, with half of them as positive pairs and half of them negative pairs. Finally, we train a random forest classifier implemented in the scikit-learn library [8] using the prepared dataset. We set the number of estimators of random forest to 200, the maximum features to 4 and the random seed to 0.

**Evaluation Protocols.** For evaluation of our method, we test trained FSL models with or without QuickBoost on standard 5-way-1-shot and 5-way-5-shot few-shot classification tasks. We use the Res12 and the Conv4 [50, 59] backbones for different FSL algorithms (i.e., ProtoNet [48], RelationNet [50], MatchingNet [53], FEAT [59], DeepSet [59], InfoPatch [34]). For evaluation of ensemble performance, we show the performance of original models before and after ensemble. All the reported accuracy scores of the original methods are based on our reproduced experimental results. The respective official implementations are used for the reproduction. For each reported number, we report the top-1 accuracy score as well as the 95% confidence interval. Note that all the results included are of the inductive set-up, while transductive set-up are of a slightly different problem formulation [3].

## 5.2 Comparison with the State-of-the-Art

Table 1 summarizes the performance of FSL-Forest in QuickBoost, original FSL model and Quickboost respectively on 5-way-1-shot and 5-way-5-shot tasks. The proposed ensemble helps trained FSL models achieve considerable performance improvement, despite the limited performance of FSL-Forest as a standalone classifier (e.g., 42.01% for 5-way-1-shot tasks on the *mini*Imagenet). For example, on Conv4-based Prototypical Network trained on the *mini*Imagenet, QuickBoost can improve its performance by 5.99% on 5-way-1-shot tasks, and 7.77% on 5-way-5-shot tasks. In general, as expected, the

improvement is more salient on 5-way-5-shot tasks compared to 5-way-1-shot tasks, which is likely because of the amplification of benefits introduced by ensemble; when the number of shot size increases, QuickBoost can return more predictions to average out bias with greater efficacy.

While FSL classification is challenging, our simplistic approach is competitive on state-of-the-art (SOTA) methods. Table 2 summarizes the SOTA methods in few-shot classification 5-way-1-shot or 5-way-5-shot tasks. Non-ensemble methods for comparison include DeepEMD [60], TADAM [40], MetaOptNet [32], LEO [45], SNAIL [38], COSOC [36] and Shot-Free [42] . Ensemble methods include Robust 20 Full [16], $E^3BM$ [35], CHEF [1], MetaOptNet+MIMO [25] and EASY [3]. As indicated, when ensembled with SOTA algorithm like InfoPatch [34], we can achieve the enhanced SOTA results on the common FSL benchmarks (e.g., 71.29% for *mini*Imagenet and 81.81% for Cifar-FS 5-way-1-shot classification). Importantly, when compared to other ensemble algorithms (e.g., Robust 20 Full [16], $E^3BM$ [35]), our approach is capable of better performance despite its little required computation resources and simplicity of usage.

## 5.3 Further Discussions

**Visualization of Ensemble Features.** We visualize the attention maps for the two different encoders, which are illustrated by Figure 3. In the figure, we have 6 pairs of attention maps. In each pair, we have, from left to right column, pairs of the original images, together with their Res12 and Res18 attention. In the first row, we have attention map comparison which is more precise when generated from a Res12 encoder, while the Res18 encoder becomes less precise in the second row. However, in the bottom-right-corner pair, the Res12 attention is inaccurate in identifying the relevant region of interest. In terms of prediction correctness, for the first row, the Res12 encoder is correct in prediction (in a 2-way-1-shot classification) while Res18 encoder is wrong, and the second row accounts for the reverse case (i.e., Res18 encoder is right and Res12 is wrong). This visualization illustrates the complementary effect among different encoders.

Note that the attention maps here are different from traditional attention maps [24, 47], and is named *mutual attention* map in this work. Instead of highlighting regions of interest in a traditional classification setting, the mutual attention map highlights regions of interest during pairwise images comparisons. Originally, attention map considers the backward-gradient-weighted features [24, 47]. Here, we visualize a set of features $\Phi(x) = \{x_i\}$ of an original image input $X$, where $i$ is the channel index from the output of an encoder $\Phi$. The output feature sum to be visualized for one particular image $x$ can be represented by $\sum_i \frac{x_i \cdot x_i'}{\|x_i\| \cdot \|x_i'\|} \cdot x_i$, where $i$ is the channel index, and $x_i'$ is per-channel-feature from the image compared.

**Encoder Combinations.** Different encoder combinations can affect ensemble results, which is summarized in Table 3. As stated in past ensemble research works [62], in general, the higher the accuracy of each model in ensemble and the difference among the ensemble components, the better the results of ensemble. This phenomenon can be reflected in the table, which are the 5-way-1-shot and 5-way-5-shot results on *mini*Imagenet, by Prototypical Networks with different encoders. Among the three different encoders, Conv4 is the most shallow network consisting of 4 CNN layers.

| Method | Backbone | *mini*Imagenet | | Cifar-FS | | *tiered*Imagenet | |
|---|---|---|---|---|---|---|---|
| | | 1-shot (%) | 5-shot (%) | 1-shot (%) | 5-shot (%) | 1-shot (%) | 5-shot (%) |
| QuickBoost | Res18 | $42.01_{\pm0.18}$ | $60.40_{\pm0.18}$ | $41.66_{\pm0.16}$ | $60.02_{\pm0.20}$ | $45.05_{\pm0.18}$ | $64.09_{\pm0.18}$ |
| ProtoNet [48] | Conv4 | $49.79_{\pm0.20}$ | $66.89_{\pm0.16}$ | - | - | | |
| **ProtoNet+***QuickBoost* | Conv4 | $\mathbf{55.78}_{\pm0.20}$ | $\mathbf{74.66}_{\pm0.16}$ | - | - | - | - |
| ProtoNet | Res12 | $60.02_{\pm0.21}$ | $75.11_{\pm0.14}$ | $66.73_{\pm0.20}$ | $77.07_{\pm0.14}$ | $68.23_{\pm0.14}$ | $84.03_{\pm0.11}$ |
| **ProtoNet+***QuickBoost* | Res12 | $\mathbf{62.30}_{\pm0.19}$ | $\mathbf{81.99}_{\pm0.14}$ | $\mathbf{71.34}_{\pm0.20}$ | $\mathbf{84.83}_{\pm0.14}$ | $\mathbf{70.40}_{\pm0.13}$ | $\mathbf{90.88}_{\pm0.11}$ |
| RelationNet [50] | Conv4 | $49.63_{\pm0.84}$ | $65.16_{\pm0.69}$ | $58.50_{\pm0.83}$ | $74.37_{\pm0.60}$ | - | - |
| **RelationNet+***QuickBoost* | Conv4 | $\mathbf{56.40}_{\pm0.89}$ | $\mathbf{71.85}_{\pm0.44}$ | $\mathbf{64.50}_{\pm0.89}$ | $\mathbf{80.67}_{\pm0.44}$ | - | - |
| MatchingNet [53] | Conv4 | - | - | $55.87_{\pm0.22}$ | - | - | - |
| **MatchingNet+***QuickBoost* | Conv4 | - | - | $\mathbf{67.60}_{\pm0.21}$ | - | - | - |
| FEAT [59] | Conv4 | $53.09_{\pm0.20}$ | $67.90_{\pm0.20}$ | - | - | - | - |
| **FEAT+***QuickBoost* | Conv4 | $\mathbf{57.08}_{\pm0.19}$ | $\mathbf{76.69}_{\pm0.18}$ | - | - | - | - |
| FEAT | Res12 | $63.50_{\pm0.20}$ | $78.35_{\pm0.16}$ | $71.76_{\pm0.20}$ | $85.14_{\pm0.15}$ | $70.80_{\pm0.14}$ | $84.79_{\pm0.11}$ |
| **FEAT+***QuickBoost* | Res12 | $\mathbf{67.70}_{\pm0.20}$ | $\mathbf{84.71}_{\pm0.15}$ | $\mathbf{74.46}_{\pm0.19}$ | $\mathbf{89.23}_{\pm0.14}$ | $\mathbf{73.32}_{\pm0.13}$ | $\mathbf{90.45}_{\pm0.11}$ |
| DeepSet [59] | Res12 | $60.34_{\pm0.20}$ | $74.47_{\pm0.16}$ | $68.04_{\pm0.18}$ | $77.26_{\pm0.15}$ | $68.59_{\pm0.14}$ | $84.36_{\pm0.11}$ |
| **DeepSet+***QuickBoost* | Res12 | $\mathbf{66.98}_{\pm0.20}$ | $\mathbf{84.00}_{\pm0.15}$ | $\mathbf{71.35}_{\pm0.17}$ | $\mathbf{82.91}_{\pm0.15}$ | $\mathbf{72.90}_{\pm0.13}$ | $\mathbf{90.44}_{\pm0.11}$ |
| InfoPatch [34] | Res12 | $67.50_{\pm0.47}$ | $82.10_{\pm0.31}$ | $79.16_{\pm0.48}$ | $89.29_{\pm0.32}$ | $71.51_{\pm0.52}$ | $85.44_{\pm0.35}$ |
| **InfoPatch+***QuickBoost* | Res12 | $\mathbf{71.29}_{\pm0.47}$ | $\mathbf{88.26}_{\pm0.33}$ | $\mathbf{81.81}_{\pm0.48}$ | $\mathbf{93.16}_{\pm0.31}$ | $\mathbf{75.47}_{\pm0.48}$ | $\mathbf{91.71}_{\pm0.34}$ |

**Table 1: Accuracy for Original Few-shot Algorithms and Ensemble.**

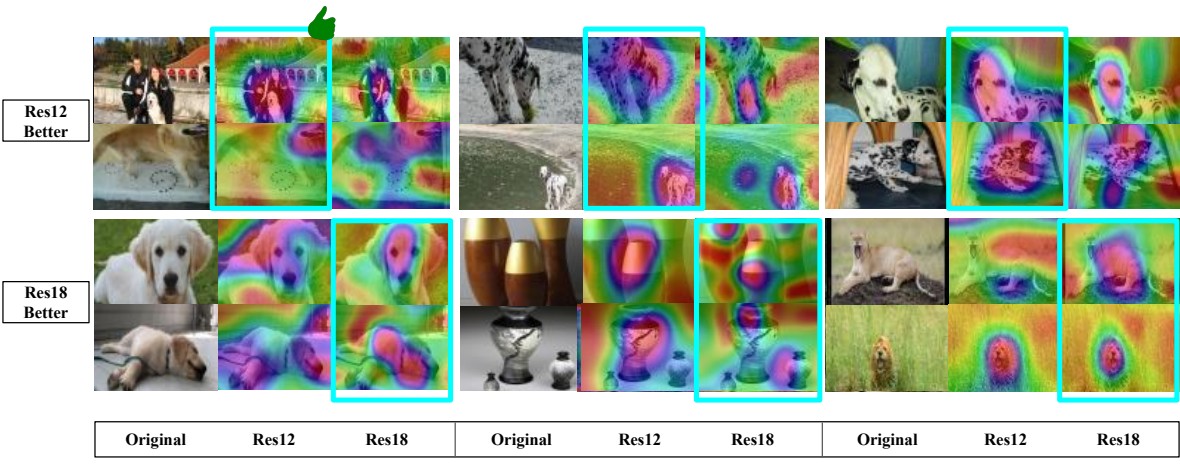

| Original | Res12 | Res18 | Original | Res12 | Res18 | Original | Res12 | Res18 |

**Figure 3: *Mutual attention* maps for 6 pairs of images. In each pair, we have, from left to right, the original image pair, their Res12 attention map and their Res18 attention map, which illustrate the complementary nature of different encoders. Best viewed in colors.**

Res12 and Res18 are similar in size (Res12 has 12 million parameters and Res18 has 11 million.). As expected, the better result (60.81% for 5-way-1-shot and 76.86% for 5-way-5-shot) belongs to the the combination of Res12 and Res18. This is because both of these two encoders are accurate on their own while possessing complementary differences against each other. Conversely, combinations involving Conv4 (e.g., Res12 + Conv4: 50.53% for 5-way-1-shot and 66.83% for 5-way-5-shot) tend to have worse performance, since Conv4-based models have a lower independent accuracy (47.79% for 5-way-1-shot tasks).

**Hyperparameter Tuning.** We study different hyperparameters for the random forest classifier of FSL-Forest on 5-way-1-shot

*mini*Imagenet classification tasks, as summarized in Table 4. We observe that FSL-Forest performance can be further improved through tuning of hyperparameters. For example, when we increase the maximum features to consider from 4 to 8 and the number of estimators from 200 to 800, the standalone FSL-Forest accuracy on 5-way-1-shot tasks increases from 42.01% to 45.17%. Besides, we observe that the classifier can be further enhanced when combined with new features like cosine similarity, which can help achieve an even higher standalone 5-way-1-shot accuracy score at 46.74%. In brief, QuickBoost has potential to achieve better performance through hyperparameter tuning and alternative designs of input features, but usually at a higher computation cost.

| Algorithm | Dataset | 5-way Accuracy (%) | |
|---|---|---|---|
| | | 1-shot | 5-shot |
| DeepEMD [60] | *mini*Imagenet | $65.91_{\pm0.82}$ | $82.41_{\pm0.56}$ |
| TADAM [40] | *mini*Imagenet | $58.50_{\pm0.30}$ | $76.70_{\pm0.30}$ |
| MetaOptNet [32] | *mini*Imagenet | $62.64_{\pm0.82}$ | $78.63_{\pm0.46}$ |
| LEO [45] | *mini*Imagenet | $61.76_{\pm0.08}$ | $77.59_{\pm0.12}$ |
| SNAIL [38] | *mini*Imagenet | $55.71_{\pm0.99}$ | $68.88_{\pm0.92}$ |
| COSOC [36] | *mini*Imagenet | $69.28_{\pm0.49}$ | $85.16_{\pm0.42}$ |
| Shot-Free [42] | *mini*Imagenet | $59.04_{\pm0.43}$ | $77.64_{\pm0.39}$ |
| Robust 20 Full [16] | *mini*Imagenet | $59.38_{\pm0.65}$ | $76.90_{\pm0.42}$ |
| $E^3BM$ [35] | *mini*Imagenet | $64.09_{\pm0.37}$ | $80.29_{\pm0.25}$ |
| CHEF [1] | *mini*Imagenet | $64.11_{\pm0.32}$ | $79.99_{\pm0.21}$ |
| MetaOptNet+MIMO [25] | *mini*Imagenet | $57.97_{\pm0.68}$ | $73.21_{\pm0.51}$ |
| EASY [3] | *mini*Imagenet | $70.63_{\pm0.20}$ | $86.28_{\pm0.12}$ |
| **InfoPatch** [34]+*QuickBoost***(Ours)** | *mini*Imagenet | $\mathbf{71.29_{\pm0.47}}$ | $\mathbf{88.26_{\pm0.33}}$ |
| Shot-Free [42] | Cifar-FS | $69.20_{\pm0.40}$ | $84.70_{\pm0.40}$ |
| EASY [3] | Cifar-FS | $75.24_{\pm0.20}$ | $88.38_{\pm0.14}$ |
| MetaOptNet+MIMO [25] | Cifar-FS | $69.99_{\pm0.73}$ | $83.71_{\pm0.48}$ |
| **InfoPatch** [34]+*QuickBoost***(Ours)** | Cifar-FS | $\mathbf{81.81_{\pm0.48}}$ | $\mathbf{93.16_{\pm0.31}}$ |

**Table 2: SOTA FSL Algorithms (Res12 Encoder).**

| Conv4 | Res12 | Res18 | 1-shot (%) | 5-shot (%) |
|---|---|---|---|---|
| Prototypical Network + *QuickBoost* | | | | |
| | ✓ | ✓ | $62.30_{\pm0.19}$ | $81.99_{\pm0.14}$ |
| ✓ | ✓ | | $55.78_{\pm0.20}$ | $74.66_{\pm0.16}$ |

**Table 3: Different encoder combination results on *mini*Imagenet.**

| Maximum Features | Number of Estimators | 1-Shot (%) | CPU Training Time (s) |
|---|---|---|---|
| 4 | 200 | 42.01 | 7.70 |
| 8 | 200 | 42.73 | 14.62 |
| 4 | 600 | 44.03 | 27.51 |
| 8 | 800 | 45.17 | 58.80 |

**Table 4: Hyperparameter Analysis for FSL-Forest.**

| Model | 1-shot (%) | Training Time ↓ | Data (# of Comparisons) ↓ |
|---|---|---|---|
| Prototypical Network (Conv4) | 49.79 | 1.1 hr | 1,500,000 |
| Prototypical Network [59] (Res12) | 60.02 | 1.5 hr | 1,500,000 |
| $E^3BM$ [35] (Res12) | 64.09 | 1.5 hr | 750,000 |
| FSL-Forest + Trained InfoPatch | 71.29 | 7.7 s | 12,800 |

**Table 5: Computation Resources Required.**

**Computation and Data Efficiency.** FSL-Forest in QuickBoost is computationally efficient. In principle, for small-scale training, traditional machine learning models such as those consisting of simple decision tree stumps can be much more computationally efficient when compared to DNNs, in terms of both time and data [54]. Table 5 summarizes the comparison between the Prototypical Network, $E^3BM$ ensemble algorithm [35] and FSL-Forest. Note that the Prototypical Network computation requirement is also the basic computation required for ensemble algorithms like Dvornik et al. [16], which involves training of multiple such DNN models. In this approximated comparison, all networks are already pretrained, and the comparison excludes the pretraining stage. The DNN is trained on one A5000 GPU, and FSL-Forest is trained on a CPU, whose model is Intel(R) Xeon(R) Silver 4310 CPU @ 2.10GHz. While the compute platforms are different (CPU vs GPU), FSL-Forest takes up negligible training time (7.7 seconds) when compared to DNN network training (approximately 1.5 hours). The numbers of comparisons are calculated based on the product of the support set class size, and the total number of queries learnt throughout the training

procedure. In particular, during training, FSL-Forest only learns from 12,800 comparisons.

**Model Interpretability.** We use Local Interpretable Model-agnostic Explanations (LIME) [43] to qualitatively understand the inner mechanisms of the model. LIME allows us to understand the effects of superpixels in test images on the prediction of class similarity. LIME scores are based on superpixels identified via unsupervised image segmentation using Felzenszwalb's graph-based image segmentation [17]. We choose LIME as it can be applied to the QuickBoost FSL-Forest classifier, which is a mixture of gradient and non-gradient based models. Furthermore, recent studies find theoretical guarantees and connections between LIME and integrated gradient methods [22]. We randomly sample two classes (class A and B) and select two images from each class (images A and B), and one image from another class (image C). We then create 2 image pairs, (i) image A and B, (ii) image A and C. We designate the randomly-selected image A as the input image for LIME's perturbations. Using a trained FSL-Forest on *mini*Imagenet, we run LIME on the model while feeding the model with pairs (i) and (ii). We find qualitative results that validate our hypothesis that the random forest classifier is able to identify image features that contribute to similarity scoring. Figure 4 shows samples of our qualitative results.

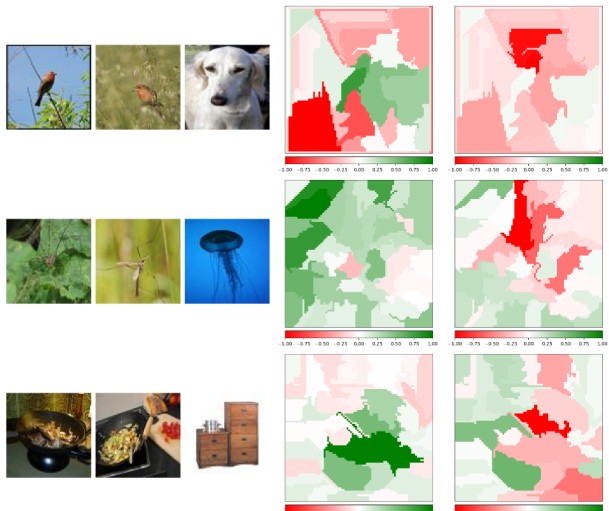

**Figure 4:** *LIME* **interpretability maps. Leftmost images are input, the middle are the same-class, and the right are the different-class. LIME maps on the left are generated based on the same-class input, and the right are the different-class. Green areas indicate regions that contributing to same class prediction, while red regions indicate the different class. Best viewed in colors.**

## 6 CONCLUSIONS

To conclude, motivated by the nature of ensemble and FSL, we explore empirically how ensemble can be effective in terms of both performance and resource. Our proposed ensemble instance strengthens the original FSL prediction scores with alternative predictions, achieving boost for FSL performance at minimal cost.

## ACKNOWLEDGMENTS

This research is supported in part by the Ministry of Education, Singapore, under its MOE AcRF TIER 3 Grant (MOE-MOET32022-0001).

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
