# OpenReview forum: "FSL-QuickBoost: Minimal-Cost Ensemble for Few-Shot Learning"
_acmmm.org/ACMMM/2024/Conference — MM2024 Poster_

### Official Review · Reviewer_wjY8 · 2024-05-21

**Rating:** 4
**Confidence:** 3

**Summary:**

This paper introduces a new ensemble method called QuickBoost for FSL to improve the generalization performances of existing FSL models with limited training data available. In particular, QuickBoost includes a deep neural network based classifier and a random forest based classifier and their output logits are averaged as the ensemble output. QuickBoost is an efficient ensemble method as the computational cost of training a random forest based classifier is much lower than that of deep models. Extensive experiments are conducted on three public FSL datasets and QuickBoost shows clear improvements compared to different baseline methods.

**Strengths:**

1.	The paper is well organized overall and is easy to follow.
2.	The proposed QuickBoost is easy to implement and could be applied to different baseline methods.
3.	The authors provide theoretical analysis which would help readers understand the benefit of ensemble methods. However, the analysis is for general ensemble methods and is not specific to the proposed method.
4.	The authors conduct experiments on three FSL datasets and QuickBoost is compared with both ensemble and non-ensemble methods to show its effectiveness. Besides, QuickBoost also clearly boosts the performances of several existing baseline methods.

**Limitations:**

1.	Several parts of this paper, i.e., (c) FSL-Forest and (d) Ensemble, are not well presented and many important details are missing. Some expressions are also confusing. For example, for each decision tree in the random forest, which feature is selected as its input? What is the actual element in $Q_m$ and is $d$ in Eq.4 a scaler or a vector? As the random forest based classifier is the main part of the proposed method, I think the meaning of these variables and the learning procedure of the random forest should be explained more clearly. Moreover, the authors do not formally explain how to obtain the logits of different classes with random forest based classifier as each decision tree only produces a single score. Besides, at line 452, how to obtain the test accuracy of the two classifiers and which part of data is used for test? Also, I feel confused about the sentence at line 455-457.
2.	Although the random forest based classifier is an essential component of the proposed method, not only the important settings about the random forest algorithm (e.g., the depth of the decision tree, the stop criterion) but also the related ablation studies are missing. It is important to illustrate the influence of the hyper-parameters of the random forest algorithm (e.g., the number of trees, the depth of trees, the number of selected features, the number of training samples, different pretrained backbones) on the ensemble performance.
3.	Although QuickBoost achieves preferable empirical results, it is still unclear what makes it a good ensemble method and why the introduction of a random forest based classifier is important. This also weakens the motivation of QuickBoost. The authors claim that different classifiers should be diverse so as to obtain a good ensemble result but the diversity is not shown appropriately in QuickBoost. For example, the authors could analyze the difference between the logits output by the neural network based classifier and the random forest based classifier with some statistical methods, which would provide a better guidance for us to select appropriate models for ensemble. I think the visualization results with only a few cases are not sufficient for explaining this point.
4.	In table 4, as the random forest based classifier is based on the image features from a trained res18 network, I think it is unfair that the training time of this res18 network is not included for comparison.

**Suitability:**

2

---

### Official Review · Reviewer_xeS1 · 2024-05-25

**Rating:** 3
**Confidence:** 2

**Summary:**

This paper introduces QuickBoost, a method designed to enhance the generalization of Few-shot Learning (FSL). QuickBoost integrates a one-vs-all binary classifier (FSL-Forest) with standard FSL models via logit-level averaging. FSL-Forest utilizes decision tree stumps to evaluate input pairs based on their feature-level elementwise value differences. Tested across three benchmarks, the proposed method demonstrates remarkable efficiency and effectiveness.

**Strengths:**

- The introduction of the ensemble random forest method to the Few-Shot Learning (FSL) task is interesting.
- The ability to train on a CPU with notable efficiency is particularly impressive.

**Limitations:**

Weakness
- The comparison in Table 2 seems biased as it incorporates the baseline method InfoPatch[31], which is already effective. Adding QuickBoost to InfoPatch does not clearly demonstrate the distinct advantages of QuickBoost.
- For clearer comparison, it would be beneficial to separate the results for miniImageNet and Cifar-FS into two distinct sections in Table 2, similar to the format used in Table 1. The current combined presentation is confusing.
- It is unclear whether the reported training time of 7.7 seconds for FSL-Forest includes only stage C or the entire FSL-QuickBoost ensemble. Clarification is needed to accurately assess the method's efficiency.
- To better validate the effectiveness of FSL-Forest in identifying relevant features, LIME results should also be applied to other FSL methods for comparative analysis.

**Suitability:**

2

---

### Official Review · Reviewer_havN · 2024-06-04

**Rating:** 6
**Confidence:** 3

**Summary:**

This work proposed a ensemble learning framework for few-shot learning (FSL). To be specific, the authors proposed to add a random-forest-based model on top of a trained standard FSL model. The added random forest is trained on pairwise feature difference generated on another feature extractor pre-trained on some base classes. While the random forest itself is a weak FSL model, it consistently improves the standard FSL model by a very significant margin on all dataset and model combinations considered in this work.

**Strengths:**

+ This work is clearly written. The methodology is described clearly with vivid illustration of the architecture and intuition (Fig. 1).
+ The proposed framework is simple and clean, thus has good potential to be widely used by the community.
+ Empirical results are very strong. The improvements over the baseline FSL models are significant.
+ The authors also provided theoretical analysis and careful ablation study, which are informative, such as the encoder combination results presented in Table 3, which successfully decoupled the effect of introducing the base encoder.

**Limitations:**

- The (d) stage in Figure 2 is kind of confiusing. It looks like the output logits by the QuickBoost module is fed as input to the standard FSL model.

**Suitability:**

3

---

### Meta-Review · Area_Chair_9H5z · 2024-06-30

**Recommendation:** Accept (Poster)
**Confidence:** 4

**Metareview:**

In this paper, the authors propose a novel ensemble method, QuickBoost, which is efficient and effective for improving the generalization of few-shot learning. After the rebuttal period, the reviewers gave consistent positive ratings. Therefore, I recommend accepting this paper. However, as Reviewers xeS1 and wjY8 pointed out, this paper still has some remaining minor concerns. The authors are required to update their paper according to the reviewers' comments and fix these issues in the final version.